# LT4/LT3 Combination Therapy vs. Monotherapy with LT4 for Persistent Symptoms of Hypothyroidism: A Systematic Review

**DOI:** 10.3390/ijms25179218

**Published:** 2024-08-25

**Authors:** Hernando Vargas-Uricoechea, Leonard Wartofsky

**Affiliations:** 1Metabolic Diseases Study Group, Department of Internal Medicine, Universidad del Cauca, Carrera 6 Nº 13N-50, Popayán 190001, Colombia; 2Medstar Health Research Institute, Georgetown University School of Medicine, Washington, DC 20007, USA; leonard.wartofsky@medstar.net

**Keywords:** hypothyroidism, persistent symptoms, levothyroxine, L-triiodothyronine, therapy

## Abstract

Regardless of the cause, hypothyroidism should be treated with levothyroxine. The objectives of management are the normalization of TSH levels and the relief of symptoms. In general, the vast majority of patients who achieve normalization of TSH levels show a resolution of symptoms; however, for a small number of individuals, symptoms persist (despite adequate control of TSH). This scenario generates a dilemma in the therapeutic approach to these patients, because even when excluding other causes or concomitant diseases that can explain the persistence of symptoms, pharmacological management strategies are scarce. Consequently, the efficacy of some less conventional approaches to therapy, such as the use of LT3 monotherapy, desiccated thyroid extracts, and LT4/LT3 combinations, in addressing persistent hypothyroid symptoms have been evaluated in multiple studies. The majority of these studies did not observe a significant benefit from these “nonconventional” therapies in comparison to results with LT4 monotherapy alone. Nevertheless, some studies report that a significant proportion of patients prefer an alternative to monotherapy with LT4. The most common approach has been to prescribe a combination of LT4 and LT3, and this review describes and analyzes the current evidence of the efficacy of LT4/LT3 combination therapy vs. LT4 monotherapy in addressing persistent hypothyroidism symptoms to provide suggested guidelines for clinicians in the management of these patients.

## 1. Introduction

Hypothyroidism is one of the most frequently occurring diseases in the world, with prevalence rates close to 10% in women and 1.5% in men, as well as incidence rates that vary according to a population’s iodine status [1,2].

By definition, hypothyroidism involves low levels of thyroid hormones (T4 and T3) in the blood, and it can be classified as primary hypothyroidism, which occurs as a consequence of the reduced function of the thyroid gland itself (e.g., due to an organ-specific autoimmune compromise (Hashimoto’s thyroiditis) or the use of radioactive iodine, exposure to ionizing radiation, antithyroid drugs, or surgery (i.e., thyroidectomy)) and secondary hypothyroidism, which occurs because of damage or functional alterations to the pituitary or hypothalamus, manifesting as an insufficiency in the production capacity of thyrotropin (TSH) or thyrotropin-releasing hormone (TRH), respectively [3,4].

The conventional treatment of the various forms of hypothyroidism is with oral levothyroxine (LT4) [5,6]. However, in the 1960s, combinations of LT4 and L-triiodothyronine (LT3) were widely used in the management of this condition under the premise that both hormones could better “simulate” the natural functioning of the gland; additionally, desiccated extracts of cow or pig thyroids were the reference products for clinical trials at that time. Interestingly, studies comparing treatments with thyroid extracts vs. LT4/LT3 combinations showed similar clinical results [7,8].

In the 1970s, it came to be understood that T4 served as a “pro-hormone” for the generation of T3 and that ~80% of T3 in peripheral tissues was derived via local conversion from T4 (a process mediated by deiodinases), and the remainder via production and secretion by the thyroid gland [7,8,9].

T3 is considered to be approximately 10-fold more potent than T4; hence, it was not surprising to learn that many patients administered LT3 are more likely to present symptoms of thyroid hormone excess. Hence, given the risks to bone (e.g., thyroid-induced osteopenia, osteoporosis, and fracture) and to the heart, the conservative management of hypothyroidism has shifted from LT4/LT3 combinations toward monotherapy with LT4 alone [8,9,10].

Although numerous studies have demonstrated the effectiveness of monotherapy with LT4 in relation to the normalization of TSH and T4 levels, a significant proportion of patients with ostensibly “normal” TSH levels experience persistent symptoms compatible with hypothyroidism, such as fatigue, weight gain, and “brain fog” [11,12]. Other complaints include cognitive alterations, excessive sleeping, cold intolerance, and, in general, decreased quality of life (QOL). Importantly, other conditions or causes of the latter symptoms (fibromyalgia, chronic fatigue, stress, depression, anemia, overweight/obesity, and micronutrient/vitamin deficiency, inter alia) have been ruled out for the most part [12,13].

Given the lack of clearcut evidence, combination therapy (LT4/LT3) for the latter group of patients has been a source of controversy over the last three decades, with inconsistent results from clinical trials and meta-analyses comparing the LT4/LT3 combination vs. LT4 monotherapy [14,15,16].

One factor against a predominant role for the use of LT3 in the treatment of hypothyroidism is its very short half-life (hours) in plasma compared to LT4 (days), which complicates dosage considerations for single daily administrations of combination formulations [17,18].

Interestingly, in 2012, guidelines by the European Thyroid Association (ETA) established that LT4/LT3 combination therapy could be used experimentally on a trial basis in patients who, although undergoing treatment with LT4 (and adequate biochemical control of TSH), experienced persistent symptoms related to hypothyroidism [19].

Although evidence based on clinical trials and meta-analyses does not support the routine use of LT4/LT3 combination therapy, some studies report that a significant proportion of patients prefer the combined therapy over monotherapy. Indeed, it is likely due to patient demand that nearly one-third of respondents in a survey of members of the American Thyroid Association [13,20] would consider LT4/LT3 combination therapy for their patients as an alternative treatment option.

This review describes and analyzes different studies that evaluated the effectiveness of combined LT4/LT3 therapy in patients with persistent symptoms of hypothyroidism and suggests a management approach for these patients.

## 2. Methods

A detailed search was carried out in the following databases: PubMed/MEDLINE, EMBASE, Scopus, BIOSIS, Web of Science, and Cochrane Library.

The search was conducted for articles published from January 1994 to June 2024 using the following keywords: primary hypothyroidism, secondary hypothyroidism, intervention (synthetic LT4/LT3 combination therapy), and LT4 (comparison).

We exclusively included clinical trials (RCTs) that evaluated the effectiveness of the LT4/LT3 combination therapy vs. monotherapy with LT4, regardless of the dose, time of use, and time of intervention, in adults (with a diagnosis of primary or central hypothyroidism).

Duplicate studies were excluded, as well as those carried out on patients with congenital or juvenile hypothyroidism and pregnant women.

In this regard, systematic reviews and meta-analyses were also reviewed and analyzed, and the studies that could be useful in accordance with the requirements were extracted.

Only studies published in English were taken into account (Figure 1).

## 3. Results

Twenty studies were found that evaluated the effect of LT4/LT3 combination therapy vs. monotherapy with LT4 on persistent symptoms in patients with hypothyroidism. The outcomes were analyzed using multiple scales or questionnaires, and the following were taken into account: mood; clinical status; QOL; cognitive function; depression; anxiety; anger; psychological distress; and physical symptoms (pain and fatigue), inter alia.

The different validated scales used for the measurement and evaluation of the described outcomes were as follows: Profile of Mood States; Visual Analogue Scale; Symptom Checklist—90; General Health Questionnaire—28; Short-Form—36 Health Survey; Thyroid Symptoms Questionnaire; ThyPRO questionnaire score; Letter–Number Sequencing working memory test; Symbol Digit Modalities; Digit Span Sub-Test of the Weschler Adult Intelligence Scale III; Trial Making Test-B; Beck Depression Inventory; Medical Outcomes Study health status questionnaire; and Comprehensive Epidemiological Screens for Depression, inter alia.

Only one study evaluated the effect of the intervention on patients with central hypothyroidism. The rest used the following inclusion criteria: individuals with primary hypothyroidism in the following scenarios: autoimmune hypothyroidism and hypothyroidism due to the use of radioactive iodine (RAI) [for Graves–Basedow disease (GBD), toxic multinodular goiter (TMG), and toxic nodule (TN)] or thyroidectomy (partial or total) for benign [multinodular goiter (MNG)] or malignant (thyroid cancer) disease. Follow-up of the intervention was carried out for a minimum of 5 and a maximum of 52 weeks, respectively.

The doses of LT3 and LT4 in the combination therapy were variable, ranging from LT4:LT3 concentrations (in µg/day) of 50:12.5, 50:10, 50:15, 50:20, 50:25, 75:5, 75:15, and 25:12.5 to doses based on the ratio percentages (5:5, 5:1, 10:1, and 1:3) and, finally, to those based on body weight (1.44 µg/kg/day for LT4 and 0.16 µg/kg/day for LT3), in addition to doses adapted according to the TSH value. In most studies, the population predominantly comprised women.

The general characteristics of the participants, design, type of intervention, and outcomes of the RCTs that evaluated the combined LT4/LT3 therapy vs. monotherapy with LT4 are summarized in Table 1.

The results of these studies suggest that, compared with LT4 monotherapy, LT4/LT3 combination therapy offers no advantages in alleviating persistent symptoms in individuals with hypothyroidism, regardless of the underlying cause. These outcomes were not affected by the measurement scales used, sex, proportions of LT4 and LT3 in the combination therapy, or time of administration of the therapy.

The effects of LT4/LT3 combination therapy (vs. LT4 monotherapy) in different outcomes for individuals with hypothyroidism are summarized in Table 2.

## 4. Discussion

Explaining the lack of a defined benefit, or “neutral” results, for many of the clinical trials described above, in relation to the alleviation of persistent symptoms in patients with hypothyroidism, is challenging. Several hypothetical explanations are proposed, as listed below.

### 4.1. Cognitive Bias

#### 4.1.1. Optimism Bias

An optimism bias is defined as the difference between a person’s expectation and the outcome that follows. If expectations are better than reality, the bias is optimistic; if reality is better than expectations, the bias is pessimistic [42].

According to this type of bias, in an individual with a base of symptoms (which are considered to be caused by hypothyroidism), any type of intervention (whether proven effective or not) can then be expected to create the perception of improvement in said symptoms. Therefore, RCTs that evaluate improvements in symptoms (based on scales or scores, see above) do not anticipate the presence of this type of bias, although crossover designs could be one way to prevent them. Because not all studies have this type of design, it is possible that patient-reported relief of symptoms may be explained (at least in part) by the presence of a cognitive bias.

#### 4.1.2. Affect Heuristic

This bias depends on the emotional responses of an individual and can occur in the cohorts studied when persons with persistent symptoms presumed secondary to existent hypothyroidism are provided with external information related to nonconventional interventions indicating that their symptoms may improve. Then, if the individuals in question find these potential benefits desirable, the patient may draw inappropriately positive or “distorted” conclusions in regard to efficacy [43].

#### 4.1.3. Authority Bias

In this type of bias, a patient may attribute greater accuracy (or truth) and knowledge to the opinion of one or more authority figures on a particular topic and, in this way, become influenced by that opinion [44].

These cognitive biases can explain (at least partially) why some patients prefer combined LT4/LT3 therapy (although the effectiveness of this intervention in symptom relief has not been demonstrated in randomized controlled clinical trials).

### 4.2. Sample Sizes of the Studies and Other Associated Biases

Because the majority of the reports to date included small sample sizes, it is likely that until studies with larger sample sizes are performed, the risk of a type 2 error will remain [45].

Additionally, it should be kept in mind that only an average of 15% of patients claim persistent symptoms while undergoing treatment with LT4, leaving the vast majority of subjects in the reported studies asymptomatic. This raises the possibility that any improvements during a trial of the combination therapy would be difficult to determine, and their lack of response would result in a “dilution of effect”. Moreover, the probability that patients with very marked symptoms may respond better to combined therapy (i.e., selection bias) could not be ruled out [45,46].

It is also possible that the durations of the treatment study periods and follow-ups with participants were not sufficient to demonstrate differences among the evaluated groups (i.e., LT4/LT3 combination therapy vs. monotherapy with LT4). In this regard, it is possible that patients with persistent symptoms might require longer durations of treatment with combined therapy to demonstrate its benefit. In such cases, a “follow-up bias” can explain the absence of differences among the interventions.

Finally, these studies cannot fully take into account confounding factors that may have affected patient responses, such as the presence of concomitant autoimmune diseases, fibromyalgia, chronic fatigue, vitamin and/or micronutrient deficiency, occult malignancy, and depression. These conditions can cause symptoms similar to hypothyroidism and, therefore, in their presence (without identification or correction), they can eventually “dilute” the effect of the intervention [45,46,47].

### 4.3. Methods of Evaluating Symptoms and Their Thresholds before and after Treatment

The impact that chronic diseases have on an individual’s daily life is a fundamental part of a comprehensive approach to their management. In many of the reported studies, a health-related QOL instrument was employed. These are subjective evaluations of the impacts of a disease and its treatment in different domains (physical, psychological, social, somatic, and well-being, inter alia), which are evaluated by the patient themselves through standardized questionnaires, better known as patient-reported outcome measures (PROMs) [48].

A large part of the PROMs applied in the different clinical trials previously described has been generic and nonspecific for thyroid disease, and did not differentiate the origin or etiology of the underlying hypothyroidism, e.g., whether based on Hashimoto’s disease or status post-thyroidectomy and, if so, whether for benign or malignant thyroid disease. Another potentially problematic issue concerns the questionnaire instruments that do not take into account variables such as the time of evolution and the severity of symptoms [49,50].

Some questionnaires assess patients’ preferences for a given intervention, whereas others establish different “thresholds” for the magnitudes of symptoms. For example, several studies used different cut-off points in the definition of “persistent symptoms”, as well as for improvements in said symptoms. Thus, there is an absence of universal criteria for defining the disease (persistent symptoms) and outcomes (improvements in symptoms). Currently, several PROMs (such as ThyPRO and ThyPRO39) are considered to be the most reliable for evaluations of this type of patient [32,51,52].

### 4.4. Low Tissue T3 Hypothesis

The concept that athyreotic patients can experience greater persistence of symptoms than those with hypothyroidism and some degree of residual thyroid tissue has been confirmed in rodent studies, in which tissue euthyroidism was only achieved with the administration of combined LT4/LT3 treatment [53].

However, extrapolating this concept to humans is much more complex, because the T4:T3 ratio in rats is 5–6:1, whereas in humans it is 13–15:1, which suggests that in humans there may be less of an impact on tissue due to a T3 deficiency [27,54,55,56].

From a clinical point of view, it has been suggested that the sole measurement of TSH during follow-up of patients with hypothyroidism undergoing LT4 replacement (and who are considered to be biochemically “controlled”) may not necessarily always reflect the euthyroid state of the individual. Hence, it has been suggested that the T4/T3 ratio could be a more reliable marker for monitoring these patients. Athyreotic subjects undergoing LT4 monotherapy will be administered higher T4/T3 ratios than control subjects. Employing the T4/T3 ratio as a marker of the clinical response to LT4 treatment, a recent cross-sectional study in patients with hypothyroidism observed that a low T3/T4 ratio was significantly associated with persistent symptoms (e.g., weight gain, cold intolerance, and skin problems). The study suggests that insufficient attention is being paid to monitoring the T3/T4 ratio as a marker for treatment efficacy rather than following TSH values alone [57].

On the basis of the above, LT3 supplementation can be considered to increase the T3/T4 ratio, thereby alleviating persistent symptoms in patients undergoing treatment with T4 (and with TSH levels within the normal range). However, no studies have definitively shown this to be the case. Moreover, an important limitation of LT3 substitution is its undesirable pharmacokinetic absorption pattern for available commercial preparations, which is reflected in a large, rapid increase in the serum concentration of T3 followed by a rapid decrease [58].

This acute “peak” after ingestion could explain why some patients experience temporary relief from their symptoms after a dose (although they report their subsequent return throughout the day). LT3 is absorbed in the proximal gastrointestinal tract, and the resultant rapid peak in its level and short half-life make it necessary to fractionate the dose two to three times a day. Multidaily dosing is associated with poor adherence due to missed doses, as well as overdosage, which translates into greater risks of adverse effects, such as osteoporosis, anxiety, tachyarrhythmias, atrial fibrillation, heart failure, and stroke, inter alia [59,60,61].

In contrast to the concerns raised over LT3 monotherapy, a recent study found that LT3 treatment improved the quality of life for women with symptoms of residual hypothyroidism who had been receiving monotherapy with LT4 or combination therapy with LT4/LT3. In this study, short-term treatment with LT3 did not induce biochemical or clinical hyperthyroidism and no adverse cardiovascular effects were observed [62]. Accordingly, in view of the conflicting data and lack of definitive results, there are currently no universal recommendations for the use of LT3 (monotherapy) in patients with persistent symptoms undergoing LT4 replacement and with TSH levels within the normal range.

The apparent and ongoing difficulties in the optimal management of patients with hypothyroidism undergoing LT4 replacement, as well as the desire to physiologically replace greater levels using an LT4/LT3 combination, have led to a renewed interest in the formulation of desiccated thyroid extract. Dried powder extracts from cow or pig thyroids first came into use in the late 1930s and were the primary form of replacement therapy until the 1970s, when synthetic forms of pure levothyroxine came on the market. Desiccated thyroid extracts (DTEs) fell out of favor because some patients found the odor of their tablets objectionable and there appeared to be an unacceptable level of variability from batch to batch. Moreover, and in spite of the high LT3 content compared to LT4, the stability and reliability of its potency were questioned [8,63]. Over the last few decades, the activity and potency of DTEs have met FDA and other international standards and, as a result, DTEs are increasingly being used, restoring their market share.

Indeed, some patients clearly prefer the use of DTEs (compared to combined LT4/LT3 therapy or monotherapy with LT4), either under the mistaken belief that they are “more natural” or because they enjoy the additional morning energy that is likely provided by the high LT3 content of the tablets. As with many of the other therapeutic trials with differing thyroid hormone preparations, the studies conducted with DTEs exhibited a high risk of selection bias and have not been reproducible (in contrast to the outcomes evaluated). For example, a recent systematic review found the evidence from different studies that evaluated the effect of DTEs (in relation to combined LT4/LT3 therapy or monotherapy with LT4) to be of low quality. Some individuals who presented with persistent symptoms (attributed to hypothyroidism) showed improvement in their symptoms with the use of DTEs (compared to those who received LT4 monotherapy), which could be due to the transiently high initial levels of serum T3 [64]. The authors concluded that the favorable interpretations of the results of DTE trials are likely flawed because of the absence of a specific control group, the placebo effect, and, in general, the absence of significant differences in outcomes, such as symptom relief or quality of life [65,66,67,68,69].

Advocates for the use of DTEs suggest that an improvement in symptoms may be seen because of their potential effect on the regulation of energy expenditure (they can modulate the activation of the PI3K pathway and mitochondrial metabolism), thus inducing weight loss. In this way, some observational studies have shown a negative correlation between weight and quality of life; therefore, in overweight or obese patients with persistent symptoms of hypothyroidism, it is possible that transient thyrotoxicosis (due to use of DTEs) may be associated with a better quality of life by facilitating weight loss. Conceivably, any intervention inducing weight loss in overweight or obese patients could be associated with an improvement in the quality of life and general well-being of these people [64,70,71,72].

These hypotheses must undoubtedly be tested in studies with adequate and robust designs.

### 4.5. Thr92Ala Polymorphism

Iodothyronine deiodinases (DIOs) are essential in the maintenance and availability of T3 in the blood and intracellularly. DIOs have three isoforms (DIO1, DIO2, and DIO3), which differ in their properties, distributions, and tissue specificities [73].

Gene expression of *DIO2* in humans is less restricted than in rats, and approximately 70% of circulating T3 is derived from the extrathyroidal conversion of T4 to T3 (catalyzed by DIO1 and DIO2). *DIO2* is upregulated in individuals with hypothyroidism and downregulated in patients with hyperthyroidism; therefore, changes in *DIO2* gene expression (or enzymatic activity) contribute to T3 homeostasis in multiple tissues [74,75]. Two single-nucleotide polymorphisms (SNPs) in the gene encoding DIO2, Thr92Ala (rs225014) and rs225015, were reported to be associated with impaired baseline measures of psychological well-being, and the presence of Thr92Ala has been linked to the reduced conversion of T4 to T3 in patients undergoing LT4 monotherapy, suggesting that combined LT4/LT3 treatment could result in better outcomes [11,76].

In patients who have undergone total thyroidectomy and are treated with LT4, heterozygous and rare homozygous patients carrying the Thr92Ala polymorphism in the *DIO2* gene exhibited reduced levels of free triiodothyronine (FT3), indicating that Thr92Ala-*DIO2* might inhibit the conversion of T4 to T3 via DIO2 [77].

This could explain why certain groups of LT4-treated hypothyroid patients experience an improved QOL when also being treated with LT3. Furthermore, the Thr92Ala polymorphism is found in the Golgi apparatus, which is abnormal, and could constitute a disease mechanism independent of T3 signaling [78].

So far, studies have shown that patients homozygous for Thr92Ala have a greater burden of symptoms related to hypothyroidism and a greater response to combined LT4/LT3 therapy than nonhomozygous patients. In addition, it has also been found that both homozygotes as nonhomozygotes experience significant psychological morbidity, suggesting that they all might have the potential for a positive response with the use of LT3 (among those whose symptoms persist despite receiving LT4) [79,80].

On the basis of early studies on these polymorphisms, hope has been raised that genetic testing, for example, for the Thr92Ala polymorphism, could lead to more effective personalized strategies for the management of hypothyroidism by offering the option of combination therapy to those patients with persistent symptoms who are receiving LT4 monotherapy and have been shown to exhibit the polymorphism [76].

Unfortunately, however, the existing data remain inconclusive. Table 3 summarizes possible hypotheses that explain the lack of effectiveness demonstrated in the studies that have evaluated the combined LT4/LT3 therapy in patients with persistent symptoms of hypothyroidism.

### 4.6. Treatment Options for Individuals with Persistent Symptoms?

The current management strategies for clinical scenarios in which individuals experience persistent hypothyroidism, despite adequate replacement with LT4 and normal TSH levels, are limited, with a variable response spectrum; therefore, they must be individualized for each case. Some approaches are summarized below:Develop a checklist for all additional or concomitant causes that may explain a patient’s symptoms (other autoimmune diseases, rheumatology, deficiencies in multiple vitamins and/or micronutrients, adequate intake and adherence to LT4, etc.).First, be aware of the sensitivity and reliability of the TSH measurement method employed for your patient and what constitutes the true target reference range during therapy. The latter may differ, for example, in patients on a replacement dosage vs. those with thyroid nodules or a history of thyroid cancer. Where such trial periods are indicated, consider the option of increasing the dose of LT4 to achieve a low–normal or low TSH value. Few studies have evaluated the effect of maintaining a low–normal or low TSH level on symptom relief in patients with hypothyroidism; some studies found an improvement in the well-being of these patients, whereas others did not. However, some patients prefer to maintain TSH values below the normal range, reporting the relief of symptoms. In this sense, weight reductions and decreases in total cholesterol and LDLc have been documented in these individuals, which may explain, at least in part, the relief of symptoms experienced [26,81,82,83].Consider the use of DTEs. Currently, natural desiccated thyroid (NDT) is used, derived from pig thyroid glands, for example, Armour Thyroid (60 mg or “one grain”), containing 9 µg of LT3 and 38 µg of LT4. The therapy is usually instituted using low doses, with increments which depend on the cardiovascular status of the patient. The usual starting dose is 30 mg/day, increasing by 15 mg every 2 to 3 weeks. Most patients require 60 to 120 mg/day. Maintenance dosages of 60 to 120 mg/day usually result in normal serum T4 and T3 levels. An adequate response to therapy is capable of normalizing TSH and T4 levels after 2 to 3 weeks [84]. However, it should be taken into account that DTEs are not available in many countries, so their use and clinical experience is very limited. In this way, the recommendation for this type of therapy must be analyzed and defined according to availability, costs, patient knowledge (benefits and risks), and, finally, by the best scientific evidence in this regard.Consider LT4/LT3 combination therapy. Studies that have evaluated LT4/LT3 combination therapy used variable doses of both LT4 and LT3 (Table 1). Therefore, if a decision is made to start combined LT4/LT3 therapy, it must be taken into account that the daily production of T3 for a 70 kg person is approximately 30 µg, of which 5 to 6 µg is secreted by the thyroid, and the rest (~25 µg/day) comes from peripheral conversion [85]. Therefore, a “physiological” replacement therapy aimed at providing the T3 that is not produced (for example, in athyrotic patients) would require minimal doses of LT3 [76,85,86]. The universally supported strategy for combination therapy is an LT3:LT4 ratio of 1:16, administered twice daily (Table 4).Consider the use of LT3 monotherapy.

As previously mentioned, evidence of the benefits of monotherapy with LT3 and its role in alleviating persistent symptoms of hypothyroidism in patients is very scarce and circumstantial; therefore, it is not recommended, and its initiation must be undertaken in consultation with the patient, explaining the potential associated risks, including greater negative cardiovascular, bone, and emotional outcomes (e.g., anxiety and psychosis), inter alia.

Table 5 describes the available commercial presentations of LT3 (monotherapy).

Finally, there have been encouraging results with other commercial presentations of LT3, such as slow-release LT3 formulations (LT3-sulfate and capsules made of poly-zinc-LT3), which have better safety, pharmacokinetic, and pharmacodynamic profiles. However, there are still no long-term studies in this clinical setting [87,88].

Figure 2 presents a proposal for a treatment approach in patients with persistent symptoms of hypothyroidism.

## 5. Conclusions

Several clinical trials, meta-analyses, and systematic reviews have evaluated the effectiveness of LT4/LT3 combination therapy vs. LT4 monotherapy in individuals with persistent symptoms of hypothyroidism. To date, a significant benefit of the combined therapy has not been demonstrated; however, multiple factors may explain this lack of effectiveness (e.g., small sample sizes of studies, biases, heterogeneity of studies, nonrobust designs, methods used to evaluate results, symptoms, genetic alterations, and confounding factors). Despite this, some patients may experience symptom improvements with combination therapy. In fact, a significant number of patients prefer combined therapy (or with DTE) vs. monotherapy with LT4; therefore, “no evidence of differences” between the LT4/LT3 combination therapy and LT4 monotherapy should not be confused with “evidence of no differences”.

Clearly, adequately designed RCTs that use precise definitions for the severity, magnitude, and time of evolution of symptoms (and outcomes) are needed, as well as methods that are sufficiently sensitive, specific, and reliable, taking into account other factors that may influence both symptoms and outcomes (e.g., genetic, concomitant health conditions, and environmental factors).

## Figures and Tables

**Figure 1 ijms-25-09218-f001:**
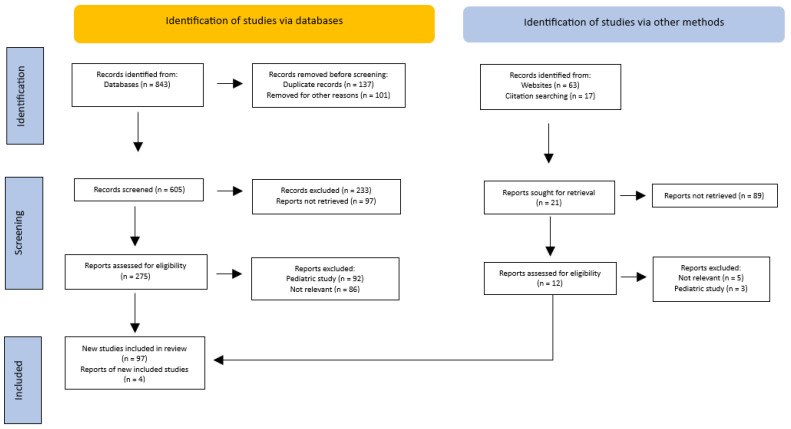
PRISMA flow diagram of the method for the selection of the articles.

**Figure 2 ijms-25-09218-f002:**
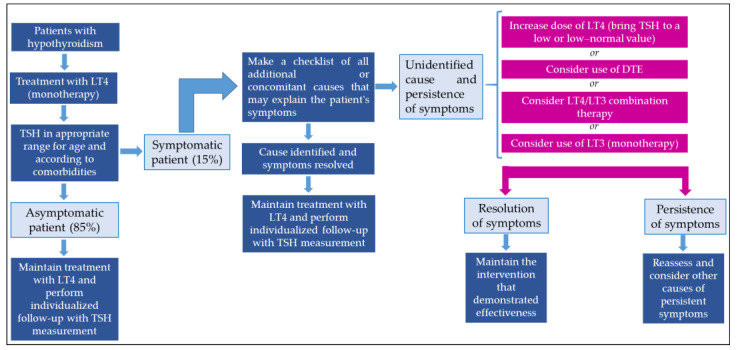
Approach to managing patients with persistent symptoms of hypothyroidism. DTE: desiccated thyroid extract.

**Table 1 ijms-25-09218-t001:** Clinical trials that evaluated the utility of LT4/LT3 combination therapy vs. LT4 in patients with persistent symptoms of hypothyroidism, regardless of the cause.

Author (Year), [Ref]	Country	Age (Mean or Range, Years)	Design	Intervention: Dose (LT4 + LT3) and Mean Duration of Treatment [Weeks]	Outcomes	N (% Women)	Diagnosis
Bunevicius (1999), [21]	Lithuania	46	Randomized/crossover	LT4, 50 µg/day; LT3, 12.5 µg/day [5]	Cognitive performance, mood, and physical symptoms	33 (93.9)	AIT or thyroid cancer (near-total thyroidectomy)
Bunevicius (2000), [22]	Lithuania	46	Randomized/crossover	LT4, 50 µg/day; LT3, 12.5 µg/day [5]	Cognitive function and psychological state	26 (100)	AIT or thyroid cancer (total thyroidectomy)
Bunevicius (2002), [23]	Lithuania	34	Randomized/crossover	LT4, 50 µg/day; LT3, 10 µg/day [5]	Mood, cognitive function, and physical symptoms	13 (100)	Subtotal thyroidectomy for GBD
Clyde (2003), [24]	USA	44.1	Clinical trial, parallel	LT4, 50 µg/day; LT3, 15 µg/day [16]	Standardized tests of neurocognitive function	46 (82)	AIT, RAI for toxic diffuse goiter, RAI for MNG, thyroidectomy for thyroid cancer or MNG
Sawka (2003), [25]	USA	49.5	Double-blind, randomized, controlled trial	LT4, 50 µg/day; LT3, 25 µg/day [15]	Mood and well-being	40 (90)	Primary hypothyroidism
Walsh (2003), [26]	Australia	47.7	Double-blind, crossover trial	LT4, 50 µg/day; LT3, 10 µg/day [10]	Cognitive function, quality of life scores, thyroid symptoms, and satisfaction with treatment	110 (93)	AITD, thyroidectomy, or RAI
Siegmund (2004), [27]	Germany	23–69	Double-blind, crossover	LT4, 5%; LT3, 5% (aim 14:1 ratio) [12]	Mood states and cognitive functioning	23 (73.9)	Thyroidectomy, RAI, or AIT
Appelhof (2005), [28]	Croatia	40	Clinical trial, parallel	LT4/LT3, 10:1, or 5:1 [15]	Mood, fatigue, psychological symptoms, and neurocognitive tests	141 (85)	AIT
Escobar-Morreale (2005), [29]	Spain	48	Double-blind, crossover trial	LT4, 75 µg/day; LT3, 5 µg/day [8]	QOL, psychometric tests, and patients’ preferences	28 (100)	AIT or RAI (for GBD or TMG)
Rodriguez (2005), [30]	USA	47.5	Double-blind, crossover	LT4, 50 µg/day; LT3, 10 µg/day [6]	Fatigue, mood, cognition, and physical symptoms	30 (89)	Autoimmune hypothyroidism, RAI, thyroidectomy
Saravanan (2005), [31]	UK	57.3	Double-blind, controlled trial	LT4, 50 µg/day; LT3, 10 µg/day [32]	Psychological well-being, thyroid symptoms, mood, cognitive behavior, and physical symptoms	584 (84)	Primary hypothyroidism
Regalbuto (2007), [33]	Italy	46.4	Clinical trial, crossover	LT3:LT4 = 3:1 [24]	Psychomotor, mnemonic and discriminative–perceptive performance, and depression and anxiety	20 (85)	Total thyroidectomy for thyroid cancer, with or without RAI
Slawik (2007), [34]	Germany	51	Double-blind, crossover	LT4, 1.44 µg/kg/day; LT3, 0.16 µg/kg/day [5]	Well-being and cognitive function	29 (27.6)	Central hypothyroidism
Nygaard (2009), [35]	Denmark	46	Double-blind, crossover	LT4, 50 µg/day; LT3, 20 or 50 µg/day [12]	QOL and depression scores	59 (93.2)	Autoimmune hypothyroidism
Valizadeh (2009), [36]	Iran	39	Clinical trial, parallel	LT4, 50 µg/day; LT3, 12.5 µg/day [16]	Social dysfunction, psychosomatic problems, anxiety, and depression	60 (80)	AIT or RAI or thyroidectomy
Fadeyev (2010), [37]	Russia	41.5	Clinical trial, parallel	LT4, 25 µg/day; LT3, 12.5 µg/day [24]	Mood, symptoms, fatigue, well-being, and neurocognitive functions	36 (100)	Primary hypothyroidism
Kaminski (2016), [38]	Brazil	42.7	Double-blind, crossover	LT4, 75 µg/day; LT3, 15 µg/day [8]	QOL, physical complaints, energy and general well-being, and mood and emotions	32 (94)	AITD
Krysiak (2018), [39]	Poland	30.5	Quasi-randomized, single-blind study	LT4:LT3 = 5:1 [24]	Depressive symptoms, sexual functioning, pain, and depression	39 (100)	Partial thyroidectomy
Krysiak (2018), [40]	Poland	35.5	Clinical trial, open-label	LT4:LT3 = 5:1 [24]	Sexual function, presence and severity of depressive symptoms	20 (0)	Primary hypothyroidism
Brigante (2024), [41]	Italy	55.9	Clinical trial, longitudinal, double-blind	Adapted according to the TSH level [24]	QOL	141 (70.92)	Total thyroidectomy for benign or malignant disease, with or without RAI

AIT: autoimmune thyroiditis; AITD: autoimmune thyroid disease; GBD: Graves–Basedow disease; MNG: multinodular goiter; QOL: quality of life; RAI: radioactive iodine; TMG: toxic multinodular goiter; TSH: Thyrotropin.

**Table 2 ijms-25-09218-t002:** Summary of the different outcomes for the combined LT4/LT3 therapy vs. monotherapy with LT4 in hypothyroidism.

Intervention	Outcome	Evidence in Favor of LT4/LT3 Combination Therapy	Evidence in Favor of Monotherapy with LT4	Neutral Effect or No Differences
LT4/LT3 combination therapy vs. monotherapy with LT4 in patients with persistent symptoms of hypothyroidism	Clinical status	No	No	Yes
	Depression	No	No	Yes
	Fatigue	No	No	Yes
	Pain	No	No	Yes
	Anxiety	No	No	Yes
	Anger	No	No	Yes
	QOL	No	No	Yes
	Psychological distress	No	No	Yes
	Mood	No	No	Yes

**Table 3 ijms-25-09218-t003:** Causes and possible explanations for the results of studies that evaluated combined LT4/LT3 therapy vs. LT4 monotherapy in patients with persistent symptoms of hypothyroidism.

Causes	Probable Explanations
Cognitive bias	Optimism bias, affect heuristic, and authority bias.
Other bias	Selection bias, dilution of the effect, and follow-up bias.
Small sample size	Low statistical power; consequently, estimates will be less precise and the probability of finding significant differences among the groups will be lower, with a greater probability for a type 2 error.
Confounding factors	Concomitant autoimmune diseases, fibromyalgia, chronic fatigue syndrome, and vitamin and/or micronutrient deficiencies.
Methods of evaluating symptoms before and after treatment	The sensitivity and specificity of the questionnaires and scales, inter alia, in the evaluation of patients probably do not accurately reflect or evaluate the present symptomatology.
Threshold of symptoms before and after treatment	There is no definition of “threshold” for the evaluation of symptoms and their severity; there is probably a wide spectrum of presentations, which could explain why some patients report improvements with combined therapy and others do not. A priori, it can be proposed that those patients with more severe symptoms will have better outcomes than those with more subtle symptoms.
TSH does not necessarily reflect tissue “euthyroidism”	Some studies suggest that a TSH level within the “normal” range does not necessarily indicate a “euthyroid” tissue state, so it would not be the best parameter for the evaluation of these patients.
Low tissue T3	It is likely that the different doses for the combined LT4/LT3 therapy used in the clinical trials do not allow for the establishment of a physiological (or almost physiological) ratio of T4/T3; therefore, it is possible that, upon reaching a ratio that is as physiological as possible, the response will also be more appropriate.
Thr92Ala polymorphism	In the presence of this polymorphism, the conversion of T4 to T3 by DIO2 is inhibited, predisposing to greater therapeutic failure with combination therapy.

**Table 4 ijms-25-09218-t004:** Commercial presentations of LT4/LT3 combination therapy.

Name	LT3 Dose	LT4 Dose
Prothyroid	10 µg	100 µg
Novothyral	5, 15, and 20 µg	25, 75, and 100 µg
Thyreotom forte	10 and 30 µg	40 and 120 µg

**Table 5 ijms-25-09218-t005:** Commercial presentations of LT3 and their doses.

Name	LT3 Dose
Cytomel	5, 25, and 50 µg
Thybon	20 and 100 µg
Tertroxin	20 µg
Liotyr	5 µg (softgel)

## Data Availability

The data that support the findings of this review are available from the corresponding author upon reasonable request.

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
