# Peer review of "LT4/LT3 Combination Therapy vs. Monotherapy with LT4 for Persistent Symptoms of Hypothyroidism: A Systematic Review"

_ijms, 2024, doi:10.3390/ijms25179218_

Round 1

Reviewer 1 Report

Comments and Suggestions for Authors

This is a review of the studies evaluating the effectiveness of combined LT4/LT3 therapy in patients with persistent symptoms of hypothyroidism. The results are clearly presented and the limitations of the available studies are adequately discussed. I do have my reservations regarding the algorithm proposed by the authors as there is no clear argument for recommending it. For instance, regarding DTE, in many countries it has been discontinued so it cannot be considered anymore; the main reason being the lack of consistency in content so I can see no reason for further recommending it. Therefore I would suggest reconsidering the algorithm or at least commenting on its limitations.

Reviewer 2 Report

Comments and Suggestions for Authors

The manuscript's content is adequate. However, some points should be edited to improve presentation.

a) The entire manuscript should be revised to avoid grammar mistakes and improve the sense of content.

b) Please check adequate use of LT3 or T3. For example, check on:

...Table 5 describes the available commercial presentations of LT3 (monotherapy)... Table 5. Commercial presentations of T3 and their doses.

c) Authors MUST do the best for quantitative comparisson; depeer discussion is required. They (and this manuscript) seem limited by personal apreciation.  

The following sentences (from conclusion section) are probe of that:

To date, a significant benefit of the combined therapy has not been demonstrated; however, multiple factors may explain this lack... Despite this, some patients may experience symptom improvements with combination therapy. In fact, a significant number of patients prefer combined therapy (or 414 with DTE) vs. monotherapy with LT4; therefore, “no evidence of differences” between the 415 LT4/LT3 combination therapy and LT4 monotherapy should not be confused with “evidence of no differences”.

d) Please check the number of references included. It should be coherent with the number of inclued studies and the additional reports included supporting the presented information.

Comments on the Quality of English Language

Use of english language is correct. However, some sentences could be edited to improve sense of content. Particular attention should be paid on the use of correct 'T3 or LT3' and T4 and LT4.
